# Impact of aerosols on ice crystal size

**Bin Zhao[1], Kuo-Nan Liou[1], Yu Gu[1], Jonathan H. Jiang[2], Qinbin Li[1], Rong Fu[1], Lei Huang[1,2], Xiaohong Liu[3], Xiangjun Shi[3], Hui Su[2], and Cenlin He[1]**

[1] Joint Institute for Regional Earth System Science and Engineering and Department of Atmospheric and Oceanic Sciences, University of California, Los Angeles, California 90095, USA.

[2] Jet propulsion Laboratory, California Institute of Technology, Pasadena, California 91109, USA.

[3] Department of Atmospheric Science, University of Wyoming, Laramie, Wyoming 82071, USA.

Correspondence to: Bin Zhao (zhaob1206@ucla.edu) and Yu Gu (gu@atmos.ucla.edu)

**Abstract.**

The interactions between aerosols and ice clouds represent one of the largest uncertainties in global radiative forcing from pre-industrial time to the present. In particular, the impact of aerosols on ice crystal effective radius ($R_{ei}$), which is a key parameter determining ice clouds' net radiative effect, is highly uncertain due to limited and conflicting observational evidence. Here we investigate the effects of aerosols on $R_{ei}$ under different meteorological conditions using 9-year satellite observations. We find that the responses of $R_{ei}$ to aerosol loadings are modulated by water vapor amount in conjunction with several other meteorological parameters. While there is a significant negative correlation between $R_{ei}$ and aerosol loading in moist conditions, consistent with the "Twomey effect" for liquid clouds, a strong positive correlation between the two occurs in dry conditions. Simulations based on a cloud parcel model suggest that water vapor modulates the relative importance of different ice nucleation modes, leading to the opposite aerosol impacts between moist and dry conditions. When ice clouds are decomposed into those generated from deep convection and formed in-situ, the water vapor modulation remains in effect for both ice cloud types, although the sensitivities of $R_{ei}$ to aerosols differ noticeably between them due to distinct formation mechanisms. The water vapor modulation can largely explain the difference in the responses of $R_{ei}$ to aerosol

loadings in various seasons. A proper representation of the water vapor modulation is essential for an accurate estimate of aerosol-cloud radiative forcing produced by ice clouds.

## 1   Introduction

Aerosols are known to interact with clouds and hence affect Earth's radiative balance, which represents the largest uncertainty in global radiative forcing from pre-industrial time to the present (IPCC, 2013). The interactions between aerosols and liquid as well as mixed-phase clouds have been extensively studied (Rosenfeld et al., 2014; Seinfeld et al., 2016; Zhao et al., 2017b), however, much less attention has been paid to ice clouds, among which cirrus clouds are globally distributed and present at all latitudes and seasons with a global cloud cover of about 30% (Wylie et al., 1994; Wylie et al., 2005). Ice clouds, formed with various types of aerosols serving as ice nucleating particles (INPs) (Murray et al., 2012; Hoose and Moehler, 2012), act as a major modulator of global radiation budget and hence climatic parameters (e.g., temperature and precipitation) by reflecting solar radiation back to space (solar albedo effect, cooling) and by absorbing and re-emitting long-wave terrestrial radiation (greenhouse effect, warming); the balance between the two is dependent on ice cloud properties, particularly ice crystal size (Liou, 2005; Waliser et al., 2009; Fu and Liou, 1993). Limited estimates (IPCC, 2013; Liu et al., 2009; Fan et al., 2016) have shown that the global aerosol-cloud radiative forcing produced by ice clouds can be very significant but highly uncertain, ranging from $-0.67$ W m$^{-2}$ to 0.70 W m$^{-2}$. For reference purposes, the best estimate of global aerosol-cloud radiative forcing produced by all cloud types is $-0.45$ W m$^{-2}$ (90% confidence interval [$-1.2$, 0 W/m$^2$]) according to the Intergovernmental Panel on Climate Change (IPCC) (Fig. TS.6 in IPCC, 2013).

The substantial uncertainty in aerosol-ice cloud radiative forcing arises largely from a poor understanding of the aerosol effects on ice cloud properties, in particular ice crystal effective radius ($R_{ei}$), a key parameter determining ice clouds' net radiative effect (Fu and Liou, 1993). Very limited observational studies (Jiang et al., 2008; Jiang et al., 2011; Su et al., 2011; Chylek et al., 2006; Massie et al., 2007) have investigated the response of $R_{ei}$ to aerosol loadings. Most of them (Jiang et al., 2008; Jiang et al., 2011; Su et al., 2011) found that polluted clouds involved smaller $R_{ei}$ than clean clouds, in agreement with the classical "Twomey effect" for liquid clouds (Twomey, 1977), which states that more aerosols can result in more and smaller cloud droplets and hence larger cloud albedo. In contrast, a couple of studies over the Indian Ocean (Chylek et al., 2006; Massie et al., 2007) reported that $R_{ei}$ is

roughly unchanged (Massie et al., 2007) or larger (Chylek et al., 2006) during more polluted episodes. It has been shown that increased aerosols (and thus INPs) lead to enhanced heterogeneous nucleation, which is associated with larger and fewer ice crystals as compared to the homogeneous nucleation counterpart (DeMott et al., 2010; Chylek et al., 2006). However, the reasons for disagreement among various studies, and the controlling factors for different aerosol indirect effects are yet to be explored, therefore the sign and magnitude of the overall aerosol effects remain in question.

With the objective to resolve the substantial uncertainty, we systematically investigate the effects of aerosols on $R_{ei}$ of two types of ice clouds under different meteorological conditions using 9-year continuous satellite observations from 2007 to 2015. The study region is East Asia and its surrounding areas (15º-55º N, 70º-135º E; Fig. S1), where aerosol loadings can range from small to extremely large values in different locations and time periods and aerosol types are varied (Wang et al., 2017; Zhao et al., 2017a).

## 2   Data and Methods

### 2.1   Sources of observational data

We obtain collocated aerosol/cloud measurements primarily from MODIS (Moderate Resolution Imaging Spectroradiometer) onboard the Aqua satellite, and CALIPSO (Cloud-Aerosol Lidar and Infrared Pathfinder Satellite Observations), as summarized in Table S1.

We acquire aerosol optical depth (AOD) retrievals at 550 nm from the level 2 MODIS aerosol product (MYD04, Collection 6) at a resolution of 10 km × 10 km. The accuracy of AOD (denoted by $\tau$) retrievals has been estimated to be about $\pm(0.05 + 0.15\tau)$ over land and $\pm(0.03 + 0.05\tau)$ over ocean (Levy et al., 2010; Remer et al., 2005). Similarly, we obtain cloud effective radius (equivalent to $R_{ei}$ in the case of ice phase) and cloud phase determined by the "cloud optical property" algorithm from the level 2 MODIS cloud product (MYD06, Collection 6) at a 1 km × 1 km resolution (Platnick et al., 2015). The MYD06 product provides an estimate of the uncertainty in $R_{ei}$ for each pixel, which takes into account a variety of error sources including 1) instrument calibration, 2) atmospheric corrections, 3) surface spectral reflectance, and 4) forward radiative transfer model, e.g., the size distribution assumption (Platnick et al., 2015). The pixel-level $R_{ei}$ uncertainties for the samples used in this study are 6.41% ± 4.97% (standard deviation). In the subsequent analysis (Section 3.1-3.3) we use mean $R_{ei}$ within certain AOD bins and the uncertainties are smaller than those for individual pixels. Also, we focus on $R_{ei}$ changes in response to aerosol loading instead of

absolute $R_{ei}$ values. For these reasons, the $R_{ei}$ uncertainty ranges are much smaller than the magnitude of $R_{ei}$ trends depicted in this study (see Figs. 1 and 3). We note that the current uncertainty evaluation has not considered the assumptions of ice crystal habit (shape), which will be discussed in Section 3.4. Stein et al. (2011) compared the MODIS $R_{ei}$ data with the "DARDAR" retrieval product (Delanoe and Hogan, 2008, 2010) based on CloudSat and CALIPSO measurements. The default DARDAR retrievals of $R_{ei}$ are mostly larger than MODIS's values, which is partly attributable to different assumptions of ice crystal habit in these two products. When the DARDAR retrievals are adjusted to mimic the MODIS assumption of ice crystal habit, the joint distribution of individual $R_{ei}$ retrievals has its peak close to the ratio of 1 between the two products, indicating a much better agreement (Stein et al., 2011). Nevertheless, the overall shape of the distributions indicates that the MODIS retrievals mostly lie between 10 and 50 μm, while the DARDAR retrievals, corrected for the crystal habit assumption, mostly lie between 10 and 80 microns. Hong and Liu (2015) reveals that the large $R_{ei}$ values in DARDAR retrievals are predominantly associated with large cloud optical thickness (> 3.0, particularly > 20). In this study, however, we focus on ice-only clouds (mostly cirrus clouds), which typically have an optical thickness less than 5.0 (see Fig. 2). For this reason, the agreement in $R_{ei}$ between MODIS and DARDAR could be better for the type of cloud used in our analysis.

The CALIPSO satellite flies behind Aqua by about 75 seconds and carries CALIOP (Cloud-Aerosol Lidar with Orthogonal Polarization), a dual-wavelength near-nadir polarization lidar (Winker et al., 2007). CALIOP has the capability to determine the global vertical distribution of aerosols and clouds. In this study, we make use of the CALIPSO level 2 merged aerosol and cloud layer product (05kmMLay, version 4.10) with an along-track resolution of 5 km and a high vertical resolution of 30-60 m below 20.2 km. The variables we employ for the investigation include aerosol/cloud layer numbers, layer base temperature, layer top/base height, layer aerosol/cloud optical depth, feature classification flags (containing the flags of "cloud type" and "aerosol type"), and two quality control (QC) flags named the cloud aerosol discrimination (CAD) score, and extinction QC (Atmospheric Science Data Center, 2012).

To examine the impact of meteorological conditions on aerosol-$R_{ei}$ relations, we also obtain vertically-resolved pressure, relative humidity (RH), and temperature from the CALIPSO aerosol profile product (05kmAPro, version 4.10), and middle cloud layer temperature ($T_{mid}$) from the CALIPSO 05kmMLay product (version 4.10). The other

meteorological parameters (see Table S1) are collected from the NCEP's Final Analysis
reanalysis data (ds083.2), which are produced at a 1º × 1º resolution every six hours. Since
Aqua and CALIPSO satellites overpass the study areas between 5:00-8:00 UTC, the ds083.2
datasets at 6:00 UTC are utilized.
## 2.2   Processing of observational data
In the analysis, we identify a CALIPSO profile layer at 5 km resolution as ice cloud when its
"cloud type" is "cirrus" or its layer base temperature is colder than –35 $^\circ$C. Previous studies
(Mace et al., 2001; Mace et al., 2006; Kramer et al., 2016) have distinguished two major types
of ice clouds characterized by distinct formation mechanisms: ice clouds generated from deep
convection (convection-generated ice clouds) and those generated in-situ due to updraft
caused by frontal systems, gravity waves, or orographic waves (in-situ ice clouds).
Considering that the impact of aerosols could differ according to formation processes, we
separate these two ice cloud types using CALIPSO data and a similar approach to that
developed by Riihimaki and McFarlane (2010). First, we group ice cloud profiles at 5 km
resolution into objects using the criteria that neighboring ice cloud profiles must vertically
overlap (the base of the higher cloud layer is lower than the top of the lower cloud layer) and
be separated by no more than 1 profiles horizontally (i.e., distance ⩽ 5 km). Only single-
layer ice cloud objects with valid QA flags (20 ⩽ CAD score ⩽ 100, Extinction QC = 0/1)
are accepted in this study. We subsequently classify ice cloud objects into three types, i.e.,
convection-generated, in-situ, and other ice clouds, according to their connection to other
clouds. The criteria to determine whether two clouds are connected are consistent with that
used to group ice cloud objects, i.e., the neighboring profiles must vertically overlap and
horizontally seperated by no more than 5 km. Convection-generated ice clouds consist of ice
cloud objects that are connected to larger clouds that include deep convective cloud profiles
(i.e., the "cloud type" flag is "deep convection"). An ice cloud object is classified as in-situ if
at least 95% of a cloud consists of a single ice cloud object which is at least 25 km (i.e., 5
profiles) in the horizontal direction, and none of the remaining profiles are deep convection
type. The remaining ice cloud objects are catogorized as the "other" type. We should be
cautious that the convection-generated and in-situ ice clouds may not be perfectly separated
using the approach described above. For example, the in-situ ice clouds indentified here could
include convectively-detrained objects that are no longer connected with their parent deep
convection, and convectively-detrained objects whose parent deep convective clouds do not

overlap with CALIPSO's track. The convection-generated ice clouds may also be contaminated by some in-situ formed ice cloud objects that happen to be spatially connected to deep convection. However, the classification scheme appears to be reasonable, as indicated by the distinct properties of the two ice cloud types shown in Section 3.2.

We then match collocated MODIS/Aqua and CALIPSO observations by averaging retrieved AOD and $R_{ei}$ from MODIS level 2 products (MYD04 and MYD06) within 30 km and 5 km radii of each 5 km ice cloud profile from CALIPSO, respectively. The averaging is done to achieve near-simultaneous aerosol and cloud measurements, since AOD observations from MODIS are missing at cloudy conditions. As AOD variation has a large spatial length scale of 40-400 km (Anderson et al., 2003), it is averaged within a larger radius than that for $R_{ei}$ to increase the number of data points with valid AOD observations. The average $R_{ei}$ is calculated based on the pixels with "cloud phase" of ice and $R_{ei}$ uncertainty smaller than 100%. Apart from the column AOD, we also need to obtain AOD of the aerosol layers mixed with ice cloud layers, as in-situ ice clouds are primarily affected by aerosols at the ice cloud height. For this purpose, we use the CALIPSO 05kmMLay product to select the aerosol layers which have valid QA flags (-100 $\leqslant$ CAD score $\leqslant$ -20, Extinction QC = 0/1; Huang et al., 2013) and are vertically less than 0.25 km away from the ice cloud layer following Costantino and Breon (2010). The AOD of these aerosol layers are averaged within a 30 km radius of ice cloud profiles. The meteorological parameters from the NCEP datasets (ds083.2) are matched to the CALIPSO resolution by determining which NCEP's grid contains a certain CALIPSO 5 km profile. Finally, we eliminate profiles with column AOD > 1.5 to reduce the potential effect of cloud contamination (Wang et al., 2015).

Convection-generated ice clouds are generated by convective updraft originating from lower troposphere and are therefore affected by aerosols at various altitudes, whereas in-situ ice clouds are primarily dependent on aerosols near the cloud height. For this reason, we use column AOD and layer AOD mixed with ice clouds as proxies for aerosols interacting with convection-generated and in-situ ice clouds, respectively. We also investigate the overall effect of aerosols on all types of ice clouds. In this case, column AOD is used as a proxy for aerosol loading affecting ice clouds following a number of previous studies (Jiang et al., 2011; Massie et al., 2007; Ou et al., 2009). The rationale is that the MODIS-detected AOD generally shows a close correlation to the MLS (Microwave Limb Sounder)-observed CO concentration in ice clouds (Jiang et al., 2008; Jiang et al., 2009), which in turn correlates well with the aerosol loading mixed with clouds in accordance with both aircraft measurements and

atmospheric modeling (Jiang et al., 2009; Li et al., 2005; Clarke and Kapustin, 2010). After the preceding screening, about $2.73 \times 10^4$, $1.09 \times 10^4$, and $5.68 \times 10^4$ profiles are used to analyze the relationships between column/layer AOD and $R_{ei}$ of convection-generated, in-situ, and all types of ice clouds. The available profiles for in-situ ice clouds are fewer because aerosols mixed with ice clouds are often optically thin or masked by clouds and hence may not be fully detected by CALIPSO.

## 2.3  Cloud parcel model simulation

To support the key findings (i.e., the water vapor modulation of $R_{ei}$-aerosol relations) from satellite observations and elucidate the underlying physical mechanisms, we perform model simulations using a cloud parcel model, which was originally developed by Shi and Liu (2016) and updated in this study to incorporate immersion nucleation. The model mimics formation and evolution of in-situ ice clouds in an adiabatically rising air parcel. The model's governing equations that describe the evolution of temperature, pressure, and mass mixing ratio, number concentration, and size of ice crystals can be found in Pruppacher and Klett (1997). The main microphysical processes considered include homogeneous nucleation and two modes of heterogeneous nucleation (deposition and immersion nucleation), depositional growth, sublimation, and sedimentation. The rate of homogeneous nucleation of supercooled sulfate droplets is calculated based on the water activity of sulfate solution (Shi and Liu, 2016). The dry sulfate aerosol is assumed to follow a lognormal size distribution with a geometric mean radius of 0.02 μm. The deposition nucleation on externally mixed dust (deposition INP) and immersion nucleation of coated dust (immersion INP) are parameterized following the work of Kuebbeler et al. (2014); the critical ice supersaturation ratios are 10% (T ≤ 220 K) or 20% (T > 220 K) for the former, and 30% for the latter. Anthropogenic INPs are not included in the cloud parcel model following recent studies (Shi and Liu, 2016; Kuebbeler et al., 2014). This is because 1) ice nucleation experiments for black carbon show contradicting results (Hoose and Moehler, 2012), and 2) ice nucleation parameterizations for anthropogenic aerosol constituents other than black carbon have not been adequately developed under ice cloud conditions due to limited experimental data. Also, we find that the relationships between $R_{ei}$ and loadings of dust aerosols are similar to those between $R_{ei}$ and loadings of all aerosols (Section 3.1). As such, we argue that the general pattern of simulation results would remain unchanged if more INPs were incorporated. The accommodation coefficient of water vapor deposition on ice crystals is assumed to be 0.1 (Shi and Liu, 2016). The sedimentation

velocity of ice crystals is parameterized following Ikawa and Saito (1991). The model
neglects some ice microphysical processes such as aggregational growth of ice crystals.
Although aggregational growth can affect the concentration and size of ice crystals, its effects
should be relatively small in terms of the response of $R_{ei}$ to aerosol loading since this process
is not strongly dependent on aerosols.
We conduct two groups of numerical experiments with different available water amount
for ice formation, denoted by initial water vapor mass mixing ratios (pv). Each group is
comprised of 100 sub-groups with initial sulfate number concentrations increasing linearly
from 5 cm$^{-3}$ to 500 cm$^{-3}$. The concentration ratios of externally mixed dust (deposition INP),
coated dust (immersion INP), and sulfate (not INP) are prescribed with values of
0.75:0.25:10000 for all experiments, since INPs represent only 1 in $10^3$ to $10^6$ of ambient
particles (Fan et al., 2016). In each sub-group, we conduct 100 one-hour experiments driven
by different vertical velocity spectra following the approach described by Shi and Liu (2016).
The vertical air motions at a 10 s resolution were retrieved from Millimeter Wave Cloud
Radar (MMCR) observations at a site located in the Southern Great Plains (SGP; 36.6°N,
97.5°W) for a 6 h period (Shi and Liu, 2016). For each of the 100 experiments, we randomly
sample a 1 h time windows from the 6 h vertical velocity retrievals, subtract the arithmetical
mean, and adjust the standard deviation to 0.25 m s$^{-1}$. The sampled vertical velocity spectra
are subsequently added a constant large-scale updraft velocity of 0.02 m s$^{-1}$ to drive the parcel
model. The initial pressure and temperature for all experiments are set at 250 hPa and 220 K,
respectively.
The model assumes that the air parcel has no mass or energy exchange with the
environment except for sedimentation of ice crystals, which is not realistic. For example, the
outburst of homogeneous nucleation in an air parcel can quickly exhaust supersaturation and
take water vapor from surrounding parcels. To conceptually mimic this process, we have
divided the 100 experiments within a sub-group into 10 combinations, each consisting of 10
experiments. It is assumed that the air parcels in the same combination can exchange water
vapor and reach equilibrium. Consequently, the occurrence of homogeneous nucleation in one
parcel will suppress the homogeneous nucleation in the connected parcels due to the depletion
of water vapor.
The ice crystal number concentration ($N_i$) and $R_{ei}$ at the end of the experiments are used to
construct the aerosol-cloud relationships. The $N_i$ for a given aerosol number concentration
(i.e., a sub-group of experiments) is calculated using an arithmetical mean of the 100
experiments, while $R_{ei}$ is calculated from mean $N_i$ and mean ice volume: $R_{ei}$ = (mean
volume/mean $N_i$ * $3/4\pi)^{1/3}$.

## 3  Results and Discussion

### 3.1  Relationships between $R_{ei}$ and aerosols modulated by meteorology

In this section we discuss the impact of aerosols on $R_{ei}$, with both ice cloud types lumped
together, based on satellite data (Fig. 1). The aerosol effects on individual ice cloud types will
be discussed in the next section. The dash line in Fig. 1a shows the overall changes in $R_{ei}$ with
AOD. $R_{ei}$ generally increases with increasing AOD for moderate AOD range (< 0.5), and
decreases slightly for higher AOD. This relationship is attributable to complex interactions
between meteorological conditions and microphysical processes, which will be detailed below.

11       Having shown overall response of $R_{ei}$ to AOD, we investigate whether the responses are

similar under different meteorological conditions. We plot the $R_{ei}$-AOD relationships
separately for different ranges of meteorological parameters, as shown in Fig. 1a-c and Fig.
S2. Included in the analysis are most meteorological parameters that can potentially affect ice
cloud formation and evolution, including the relative humidity averaged between 100 hPa and
440 hPa ($RH_{100-440hPa}$), convective available potential energy (CAPE) which is an indicator of
convective strength, middle cloud layer temperature ($T_{mid}$), wind speed and direction at ice
cloud height and at surface, vertical velocity below and at ice cloud height, and vertical wind
shear. For some meteorological parameters, e.g., vertical wind shear and vertical velocity at
300/500 hPa, the curve shapes are similar for different meteorological ranges. However, for
$RH_{100-440hPa}$, CAPE, and U-component of wind speed at 200 hPa (U200), the curve shapes
vary significantly according to different ranges (Fig. 1a-c). As illustrated by $RH_{100-440hPa}$ and
CAPE, $R_{ei}$ decreases significantly with increasing AOD for high $RH_{100-440hPa}$ (> 65%) or
CAPE (> 500 J/kg) following the rule of "Twomey effect". In contrast, for low $RH_{100-440hPa}$ (<
45%) or CAPE (0 J/kg), $R_{ei}$ generally increases sharply with AOD; an exception is that at a
large AOD range (> 0.5), a further increase in AOD could decrease $R_{ei}$ slightly. To the best of
our knowledge, the strong dependency of $R_{ei}$-AOD relationships on meteorological conditions
for ice clouds has been demonstrated for the first time.

29       These correlations, however, may not be necessarily attributed to aerosols. It is

theoretically possible that certain meteorological parameters lead to simultaneous changes in
both AOD and ice cloud properties and produce a correlation between these two parameters.
To rule out this possibility, we examine the responses of AOD to the above-mentioned

meteorological parameters (Fig. S3) and find that AOD does not serve as proxy for them since it varies by less than 0.2 in response to variation in any meteorological parameter. Furthermore, we bin observed $R_{ei}$ according to $RH_{100-440hPa}$, CAPE, and U200, for different ranges of AOD (Fig. 1d-f). Using $RH_{100-440hPa}$ as an example, a larger AOD corresponds to smaller $R_{ei}$ for a given $RH_{100-440hPa}$ within the larger $RH_{100-440hPa}$ range, whereas an increase in AOD enlarges $R_{ei}$ for a given $RH_{100-440hPa}$ within the smaller $RH_{100-440hPa}$ range. Similar results are found for CAPE and U200 (Fig. 1d-f), demonstrating the role of aerosols in altering $R_{ei}$ under the same meteorological conditions. Moreover, the cloud contamination in AOD retrieval (Kaufman et al., 2005) or aerosol contamination in cloud retrieval (Brennan et al., 2005) is not likely to lead to observed $R_{ei}$-AOD correlations, because the retrieval biases cannot explain the opposite correlations under different meteorological conditions. Therefore, we conclude that both the positive and negative correlations between AOD and $R_{ei}$ are primarily attributed to the aerosol effect. This causality is also supported by numerical simulations using a cloud parcel model to be described in Section 3.4. Furthermore, we find that the three meteorological parameters which pose the strongest impact on $R_{ei}$-AOD relationships ($RH_{100-440hPa}$, CAPE, and U200) are closely correlated with each other, with correlation coefficients between each two exceeding ±0.5 and p-value less than 0.01 (Table S2). In fact, all these three parameters are closely related to the amount of water vapor available for ice cloud formation. It is obvious that $RH_{100-440hPa}$ is an indicator of water vapor amount. CAPE represents convective strength and hence water vapor lifted to ice cloud heights; U200 is the zonal wind at 200 hPa as opposed to the meridional wind, and denotes the origin of air mass such as moist Pacific Ocean (negative U200, easterly wind) or dry inland continent (positive U200, westerly wind). Therefore, water vapor amount is likely a key factor which modulates the observed impact of aerosols on $R_{ei}$.

The proposed mechanism for the water vapor modulation is that different water vapor amount substantially alters the relative significance of different ice nucleation modes, thereby resulting in different $R_{ei}$-AOD relationships. Specifically, ice crystals form via two primary pathways: homogeneous nucleation of liquid cloud droplets (or supercooled solution particles) below about −35 ℃, and heterogeneous nucleation triggered by INPs (IPCC, 2013; DeMott et al., 2010). INPs possess surface properties favorable to lowering the ice supersaturation ratio required for freezing (IPCC, 2013; DeMott et al., 2010), therefore the onset of heterogeneous nucleation is generally easier and earlier in rising air parcels. Under moist conditions (high $RH_{100-440hPa}$, high CAPE, or negative U200), an air parcel could experience longer time for

supersaturation development, increasing the odds of exceeding the supersaturation threshold for homogeneous ice nucleation. Therefore, homogeneous nucleation dominates in this case, and more aerosols could give rise to more numerous and smaller ice crystals, which is in connection with the "Twomey effect" for liquid clouds. Under dry conditions, however, the earlier onset of heterogeneous nucleation can strongly compete with and possibly prevent homogeneous nucleation involving more abundant liquid droplets or solution particles (IPCC, 2013; DeMott et al., 2010). Therefore, more aerosols (and hence more INPs) are expected to lead to a higher fraction of ice crystals produced by heterogeneous nucleation comprising of fewer and larger ice crystals. This is known as "negative Twomey effect" as first described by Karcher and Lohmann (2003). At very large AOD range (> 0.5), heteorogeneous nucleation dominates and a further increase in aerosols would decrease $R_{ei}$ due to the formation of more smaller ice crystals. These proposed mechanisms will be supported and elaborated on using model simulations in Section 3.4.

Here an inherent assumption is that INP concentration is roughly proportional to, or at least positively correlated with AOD. Considering that INPs only account for a small fraction of ambient aerosols, we may not take this assumption for granted. Here we plot the $R_{ei}$-AOD relations using only the cases in which the "aerosol type" (a flag contained in the feature classification flags of CALIPSO) is dust (Fig. 1g-i), and find that the water modulation effect is very similar to the preceding results (i.e., Fig. 1a-c). In addition to column AOD, we also find similar dependences of $R_{ei}$ on layer AOD (mixed with in-situ ice clouds) for all aerosols and for dust only (see Fig. 3d-i). Since specific components of dust aerosols have been known as effective INPs (Murray et al., 2012; Hoose and Moehler, 2012), the similar $R_{ei}$-AOD relations of dust and of all aerosols to some extent support the proposed mechanisms for water vapor modulation.

## 3.2 $R_{ei}$-aerosol relationships for two types of ice clouds

Considering that distinct formation mechanisms of convection-generated and in-situ ice clouds may lead to different aerosol effects, we distinguish these two ice cloud types based on their connection to deep convection (Section 2.2). In the study region, the convection-generated, in-situ, and other ice clouds account for 44.9%, 52.4%, and 2.7% of all ice cloud profiles, respectively. Figure 2 illustrates the accumulative probability distribution of cloud thickness, cloud optical thickness (COT), and $R_{ei}$ of the two ice cloud types. The cloud thickness and COT of convection-generated ice clouds are remarkably larger than those of in-situ ice clouds, because more water is transported to upper troposphere in the formation

process of the former type, consistent with numerous aircraft measurement results (e.g., Kramer et al., 2016; Luebke et al., 2016; Muhlbauer et al., 2014). The $R_{ei}$ of convection-generated ice clouds is slightly larger than that of in-situ ice clouds, which has also been reported in a number of aircraft campaigns (Luebke et al., 2016; Kramer et al., 2016). The larger $R_{ei}$ in convection-generated ice clouds is attributed to larger water amount and the fact that they are produced by convection emerging from lower altitude. Below the $-35\ ^{o}C$ isotherm, ice crystals stem only from heterogeneous nucleation, which tends to produce larger ice crystals compared to the homogeneous nucleation counterpart (Luebke et al., 2016).

Figures 3 shows the impact of aerosols on $R_{ei}$ under different meteorological conditions for convection-generated and in-situ ice clouds, respectively. As described in Section 2.2, we use column AOD and layer AOD mixed with ice clouds as proxies of aerosols interacting with convection-generated and in-situ ice clouds, respectively. The most impressive feature from these figures is that the meteorology modulation remains in effect for either of the two ice cloud types, such that $R_{ei}$ generally decreases with AOD under high $RH_{100-440hPa}$/high CAPE/negative U200 conditions, whereas the reverse is true under low $RH_{100-440hPa}$/low CAPE/positive U200 conditions. Similar to the Section 3.1, we also demonstrate that the $R_{ei}$-aerosol relationships are primarily attributed to the aerosol effect by illustrating role of aerosols in altering $R_{ei}$ under the nearly constant meteorological conditions (Fig. S4). For example, a larger AOD is associated with a smaller $R_{ei}$ for a given $RH_{100-440hPa}$ within the larger $RH_{100-440hPa}$ range, while an increase in AOD leads to a larger $R_{ei}$ for a given $RH_{100-440hPa}$ within the smaller $RH_{100-440hPa}$ range. These results illustrate that the meterology modulation of aerosol effects on $R_{ei}$ is valid regardless of ice cloud formation machanisms.

A closer look at Fig. 3 shows that there exist noted differences between the $R_{ei}$-aerosol relationships for the two ice cloud types. For convection-generated ice clouds, a weak negative correlation (but still statistically significant at the 0.01 level) between $R_{ei}$ and AOD is found under moist conditions, while a strong positive correlation is found under dry conditions. Note that at a large AOD range ($> 0.5$) under dry conditions, a further increase in AOD could slightly reduce $R_{ei}$ because of the "Twomey effect" when heterogeneous nucleation prevails. For in-situ ice clouds, however, weaker positive and stronger negative correlations are shown under dry and moist conditions, respectively. As a result, overall $R_{ei}$ slightly increases with aerosol loading for convection-generated ice clouds, but slightly decreases for in-situ clouds.

These differences are again linked to the distinct formation mechanisms of the two ice cloud types. As the formation mechanism of convection-generated ice clouds is quite complex, we first briefly review major pathways of ice crystal formation in convection-generated clouds. On one hand, ice crystals are produced by heterogeneous freezing of liquid droplets at temperatures larger than about –35 $^{\circ}$C or possibly by homogeneous freezing of liquid droplets at about –35 $^{\circ}$C (Kramer et al., 2016). The ice crystals are then lifted to the temperature range < –35 $^{\circ}$C and are considered to be ice clouds (Kramer et al., 2016). On the other hand, an additional freezing of solution particles (in contrast to liquid droplets in the former case) may occur in the presence of "preexisting ice" if the updraft is sufficiently strong. The freezing mechanism is likely homogeneous nucleation, since INPs have already been consumed (Kramer et al., 2016). Such additional freezing events are very difficult to occur and hence make less important contributions to ice crystal budget (Luebke et al., 2016), since the pre-existing ice suppresses supersaturation and prevents the threshold for homogeneous nucleation to take place (Shi et al., 2015). In this study, "homogeneous nucleation" refers to freezing of liquid droplets near the –35 $^{\circ}$C isotherm as well as the freezing of solution particles below –35 $^{\circ}$C. The former could be important for ice formation, because any liquid droplets would be homogeneously nucleated when they are lifted to the –35 $^{\circ}$C isotherm. Evidence for homogeneous droplet freezing has been frequently observed in deep convective clouds and convection-generated cirrus clouds (Twohy and Poellot, 2005; Heymsfield et al., 2005; Rosenfeld and Woodley, 2000; Choi et al., 2010). In particular, liquid droplets are frequently observed to supercool to temperatures approaching –35 $^{\circ}$C and even below, and at slightly colder temperature only ice is found, which serves as strong evidence for homogeneous droplet freezing (Rosenfeld and Woodley, 2000; Choi et al., 2010). Even if the occurrence frequency of homogeneous droplet freezing is low, its contribution to ice number concentration and $R_{ei}$ may still be substantial in view of the fact that numerous ice crystals can be produced in a single homogeneous nucleation event.

Obviously, convection-generated ice clouds are influenced by aerosols at various heights, which presumably contain much more INPs than the thin upper tropospheric aerosol layers in the case of in-situ ice clouds. In addition, the heterogeneously formed ice crystals in convective clouds are able to grow before being lifted to –35 $^{\circ}$C isotherm where homogeneous nucleation bursts, giving rise to a larger difference between the ice crystal sizes produced by heterogeneous and homogeneous nucleation as compared to in-situ ice clouds. For these reasons, under dry conditions, the increase in $R_{ei}$ with aerosol loading, which is due

to the transition from homogeneous-dominated to heterogeneous-dominated regimes, would
be much more pronounced for convection-generated ice clouds.
At moist conditions, homogeneous nucleation could dominate for both ice cloud types as
described in Section 3.1, but the mass fraction of homogeneously formed ice crystals is
smaller for convection-generated ice clouds than that for in-situ ice clouds, leading to a
weaker decline in $R_{ei}$ with aerosols. Alternatively, for convection-generated ice clouds, ice
multiplication, a microphysical process in which collision between ice particles and large
supercooled droplets rapidly produces many secondary ice particles in strong updrafts
(Lawson et al., 2015; Koenig, 1965, 1963), could also play a remarkable role in ice formation.
Its role could be important only under moist conditions where cloud droplets may grow to
large sizes required for ice multiplication (Lawson et al., 2015; Koenig, 1965, 1963). The
onset of ice multiplication may suppress or even prevent homogeneous nucleation to occur. In
the situation dominated by ice multiplication, the relatively flat response of $R_{ei}$ to AOD in
case of convection-generated ice clouds can also be explained, since ice multiplication is
supposed to be stronger at the lower AOD which favors the formation of large cloud droplets.
Whether the ice formation under moist conditions is dominated by homogeneous nucleation
or ice multiplication is clearly dependent on environmental conditions such as updraft
velocity, water vapor, cloud height and thickness, etc, a subject requiring further research.

### 3.3   Seasonal variations in $R_{ei}$-aerosol relationships

Furthermore, we find that the meteorological modulation can largely explain differences in
$R_{ei}$-AOD relationships as a function of season. Figure 4a shows that the $R_{ei}$-AOD
relationships are dramatically different associated with season, such that $R_{ei}$ decreases
significantly with increasing AOD in summer (June, July, and August), while $R_{ei}$ increases
rapidly in winter (December, January, and February). Figure 4d-f illustrate the probability
distribution functions (PDFs) of $RH_{100-440hPa}$, CAPE, and U200 in different seasons (the area
under any PDF equals 1.0). The overlapping area of PDFs in summer and winter represents
the degree of difference in meteorological conditions between these two seasons. We find that
meteorological conditions are significantly distinct in summer and winter in terms of $RH_{100-}$
$_{440hPa}$, CAPE, and U200, as indicated by relatively small overlapping areas (<0.6) for these
three parameters. The $RH_{100-440hPa}$ and CAPE tend to be higher and U200 tends to be more
negative in summer. Moreover, the shapes of $R_{ei}$-AOD curves in summer and winter highly
resemble those under high-$RH_{100-440hPa}$/high-CAPE/negative-U200 and low-$RH_{100-440hPa}$/low-
CAPE/positive-U200 conditions, respectively (see Fig. 1a-c), which demonstrates that the
discrepancy in meteorological conditions between winter and summer can, to a large extent,
explain the distinct $R_{ei}$-AOD relationships in these two seasons.
With regard to different ice cloud types, the percentages of ice cloud profiles that are
convection-generated type are 38.2%, 48.1%, 51.4%, and 39.1% in winter, spring, summer,
and fall, respectively. The corresponding percentages for in-situ ice clouds are 57.0%, 49.6%,
47.0%, and 58.2%, respectively. Fig. 4b-c show that, for both ice cloud types, the $R_{ei}$-aerosol
curves in summer and winter are largely similar to those under moist and dry conditons (Fig.
3), indicating that the seasonal variations in $R_{ei}$-aerosol relations for both ice cloud types are
largely attributable to the meteorology modulation. For convection-generated ice clouds, in
winter, spring and fall, $R_{ei}$ generally increases when AOD < 0.5, characteristic of
homogeneous nucleation being overtaken by heterogeneous nucleation, while $R_{ei}$ decreases
slightly when AOD > 0.5 in accordance with heterogeneous nucleation and increasing INP
concentrations. In summer, $R_{ei}$ shows a weak decreasing trend with AOD, which could be
explained by the domination of homogeneous nucleation or ice multiplication as described in
Section 3.2. For in-situ ice clouds, a sharp decline in $R_{ei}$ with AOD is observed in summer,
attributed to the "Twomey effect" when homogeneous nucleation prevails. The trends in other
seasons are rather weak (although an increase is noticed in winter at low layer AOD). A
probable reason is that each season consists of varying meteorological conditions (Fig. 4d-f).
As shown in Fig. 3d-f, the decreasing trends in $R_{ei}$ under moist conditions are strong, while
the increasing trends under dry conditions are relatively weak. Even if the occurrence
frequency of dry conditions is large in a season, say winter, the integration of all
meteorological conditions may still yield a relative flat $R_{ei}$-aerosol relationship. Another
possible reason is that the correlation of INP concentration and layer AOD could be weak in
some physical conditions.
## 3.4  Modeling support for the water vapor modulation
We have shown that the $R_{ei}$-aerosol relationships are modulated by meteorological conditions,
particularly water vapor amount. To support the observed relationships and our proposed
physical mechanisms, we perform model simulations as described in Section 2.3 and
summarize the results in Fig. 5.
Figure. 5a reveals that the simulated patterns of $R_{ei}$-aerosol relationships under different
water vapor amount agree fairly well with the corresponding observed patterns (Fig. 1a-c).
Specifically, with an adequate water vapor supply (pv = 103 ppm), $R_{ei}$ decreases significantly
with aerosol concentrations ("Twomey effect"). Under a dry condition (pv = 78 ppm), $R_{ei}$
increases noticeably with small-to-moderate aerosol concentrations ("negative Twomey
effect"), and decreases slightly with further aerosol increase. A deeper analysis of the
simulation results supports our proposed mechanism (Section 3.1) that the competition
between different ice nucleation modes is the key to explain the water vapor modulation.
With an adequate water vapor supply (pv = 103 ppm), the onset of deposition and immersion
nucleation consumes only a small fraction of water vapor due to the small INP population.
Considerable supersaturation remains. After further updraft movement, homogeneous
nucleation is triggered and occurs spontaneously over a higher and narrow ice supersaturation
range (140-160%). Therefore, homogeneous nucleation acts as the dominant ice formation
pathway, as indicated by the very small number fraction (< 10%) of heterogeneously formed
ice crystals, shown in Fig. 5b. In this case, more aerosols are associated with the formation of
more numerous and smaller ice crystals, consistent with the simulation results of Liu and
Penner (2005). With an inadequate water vapor supply (pv = 78 ppm), Fig. 5b reveals that the
number fraction of heterogeneously formed ice crystals increases dramatically from < 1% to
~95% when aerosol number concentrations increase from 5 $cm^{-3}$ to ~300 $cm^{-3}$ (the INP
number concentrations increase proportionally). Obviously, the occurrence of heterogeneous
nucleation could consume a considerable fraction of water vapor such that the remaining
supersaturation is quite low and would require extremely strong updraft to uphold the
homogeneous nucleation threshold. When aerosol loading increases, homogeneous nucleation
is gradually suppressed and reduced to a minimum. Since the outburst of homogeneous
nucleation generally produces more ice crystals at smaller size compared with the
heterogeneous counterpart, an increasing fraction of heterogeneous nucleation would result in
fewer ice crystals with larger average size ("negative Twomey effect"). At larger aerosol
loading (300-500 $cm^{-3}$), a further aerosol increase slightly reduces $R_{ei}$ in accordance with
heterogeneous nucleation and the "Twomey effect" (all INPs are consumed in this aerosol
concentration range).
The current cloud parcel model simulates the environmental conditions and physical
processes for in-situ ice clouds. For convection-generated ice clouds, the competition between
homogeneous and heterogeneous nucleation may explain the observed $R_{ei}$-aerosol relations
especially at dry conditions; however, the formation of this ice cloud type involves additional
complex physical processes. As described in Section 3.2, ice multiplication together with
heterogeneous nucleation may play an important role and dominate the ice formation in moist
conditions. Furthermore, ice crystals in convection-generated ice clouds are formed primarily
by freezing of liquid droplets rather than nucleation on solution particles. The simulation of
the aerosol impact on convection-generated ice clouds calls for more sophisticated models
and future investigations.
As a simplified model, the simulation results of the cloud parcel model may not be
quantitatively compard with the observational data. In satellite data analysis, we used
column/layer AOD and $RH_{100\text{-}440hPa}$ (or CAPE, U200) as proxies for aerosol loading related to
ice clouds and overall available water amount at the upper atmosphere, respectively.
However, the cloud parcel model only tracks the aerosol number concentration and water
vapor within a single air parcel. It is clear that a direct and quantitative comparison between
satellite observations and model results requires developing a 3-D atmospheric model and
analysis, a difficult task for further investigation in the future. Although the indices are not
exactly the same, we submit that the simulated dependency of $R_{ei}$ on aerosols could be used to
qualitatively interpret the observed relationships, because the indices used in satellite analysis
(AOD and $RH_{100\text{-}440hPa}$) and parcel model (aerosol number concentration and water vapor
mixing ratio) are closely correlated with each other, and that the meteorological parameters
and aerosol concentration ranges used in the simulations are representative of typical in-situ
ice clouds.
Finally, a factor that could potentially induce changes in satellite-retrieved $R_{ei}$ but has not
been considered is the habit of ice crystals. Based on previous studies (Bailey and Hallett,
2009; Lawson et al., 2006; Lynch et al., 2002), the habit of ice crystals is dependent on a
number of factors, among which the most important one is temperature, followed by ice
supersaturation ratio. In this study we focus on $R_{ei}$ changes with aerosol loading, for which
temperature does not appear to have noticeable effect. For supersaturation ratio, the formation
of ice crystals under moist conditions is dominated by homogeneous nucleation, therefore the
ice supersaturation ratio surrounding ice crystals is usually very low and the ice habit is not
likely to change significantly with aerosol loading. Under drier conditions, however,
heterogeneous nucleation gradually takes over homogeneous nucleation with aerosol loading
increase. Subsequently, the supersaturation ratio surrounding ice crystals would become
higher, possibly leading to changes in ice crystal habit. Considering that a single habit (i.e.,
aggregated column) is assumed in Collection 6 MODIS retrieval algorithm (Platnick et al.,
2015), ice habit changes could possibly induce changes in the satellite-retrieved $R_{ei}$. However,
this retrieval bias should not change our major conclusion about the aerosol impact on ice
crystal size, which has been supported by the cloud parcel modeling used in this study. The

quantitative assessment of the impact of ice crystal habit on satellite retrievals of $R_{ei}$ is a very complicated and difficult task that merits further study.

## 4   Conclusions and implications

In this study, we investigate the effects of aerosols on $R_{ei}$ under different meteorological conditions using 9-year satellite observations. We find that the responses of $R_{ei}$ to aerosol loadings are modulated by water vapor amount in conjunction with several other meteorological parameters, and vary from a significant negative correlation ("Twomey effect") to a strong positive correlation ("negative Twomey effect"). Simulations using a cloud parcel model indicate that the water vapor modulation works primarily by altering the relative importance of different ice nucleation modes. The water vapor modulation holds true for both convection-generated and in-situ ice clouds, though the sensitivities of $R_{ei}$ to aerosols differ noticeably between these two ice cloud types due to distinct formation mechanisms. The water vapor modulation can largely explain the different responses of $R_{ei}$ to aerosol loadings in various seasons.

$R_{ei}$ is a key parameter determining the relative significance of the solar albedo (cooling) effect and the infrared greenhouse (warming) effect of ice clouds; the variation of $R_{ei}$ could change the sign of ice clouds' net radiative effect (Fu and Liou, 1993). Aerosols have strong and intricate effects on $R_{ei}$ through their indirect effect. We provide the first and direct evidence that the competition between the "Twomey effect" and "negative Twomey effect" is controlled by certain meteorological parameters, primarily water vapor amount. Consequently, the first aerosol indirect forcing, defined as the radiative forcing due to aerosol-induced changes in $R_{ei}$ under a constant ice water content (IPCC, 2013; Penner et al., 2011), would change from positive to negative between high and low RH ranges, implying that the water vapor modulation could play an important role in determining the sign, magnitude, and probably seasonal and regional variations of aerosol-ice cloud radiative forcings. An adequate and accurate representation of this modulation in climate models will undoubtedly induce changes in the magnitude and sign of the current estimate of aerosol-ice cloud radiative forcing. Finally, although this study focuses on East Asia, we anticipate that the present findings might be generalized to other regions as well in view of the fact that the aerosol loadings in East Asia usually span a larger range than other regions due to substantial emissions (Zhao et al., 2017a; Wang et al., 2014) and that the aerosol effects on ice cloud properties are particularly pronounced at low and moderate aerosol loadings (Figs. 1, 3, 4).

## Acknowledgements

Research work contained in this paper has been supported by NSF EAGER Grant AGS-1523296, NSF Grant AGS-1701526, and NASA ROSES ACMAP and CCST grants. We also acknowledge the support of the Joint Institute for Regional Earth System Science and Engineering at University of California, Los Angeles and the Jet Propulsion Laboratory, California Institute of Technology, under contract with NASA.

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

**Figures**

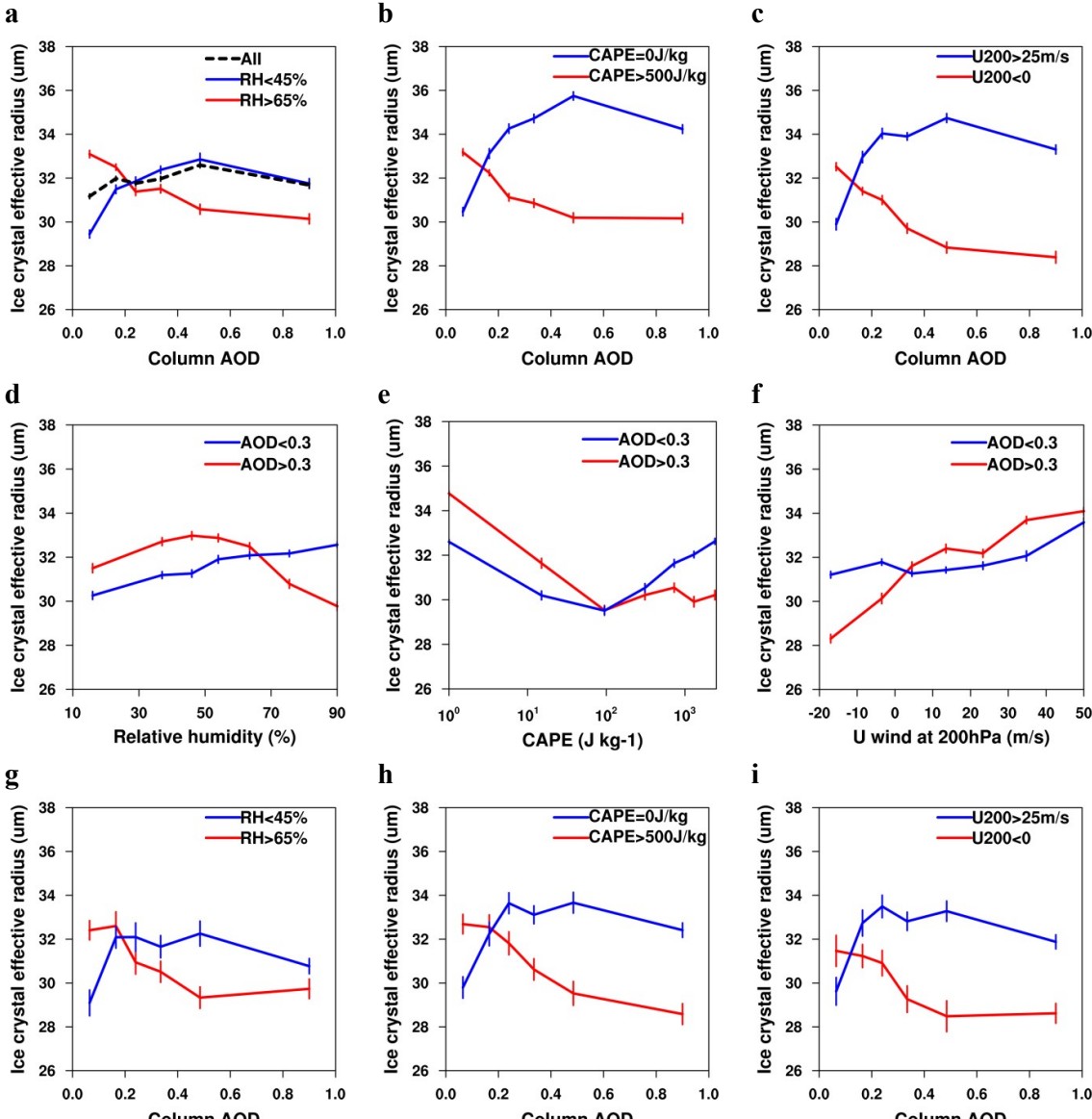

Figure 1. Influence of aerosols on ice crystal effective radius ($R_{ei}$) of ice clouds modulated by
meteorological conditions. (a-c) Changes in $R_{ei}$ with AOD for different ranges of (a) $RH_{100-440hPa}$,
$_{440hPa}$, (b) CAPE, and (c) U200. (d-f) Changes in $R_{ei}$ with (d) $RH_{100-440hPa}$, (e) CAPE, and (f)
U200 for different ranges of AOD. (g-i) The same as (a-c) but for the profiles with dust
aerosols only. The meteorological parameters and AOD are divided into 3 and 2 ranges
containing similar numbers of data points, respectively; the curves for the medium
meteorological range are not shown. The error bars denote the standard errors ($\sigma/\sqrt{N}$) of the
bin average, where $\sigma$ is the standard deviation and N is the sample number. The influences of
other meteorological parameters are shown in Fig. S2. The total number of samples used in
this figure is $5.68\times10^4$.

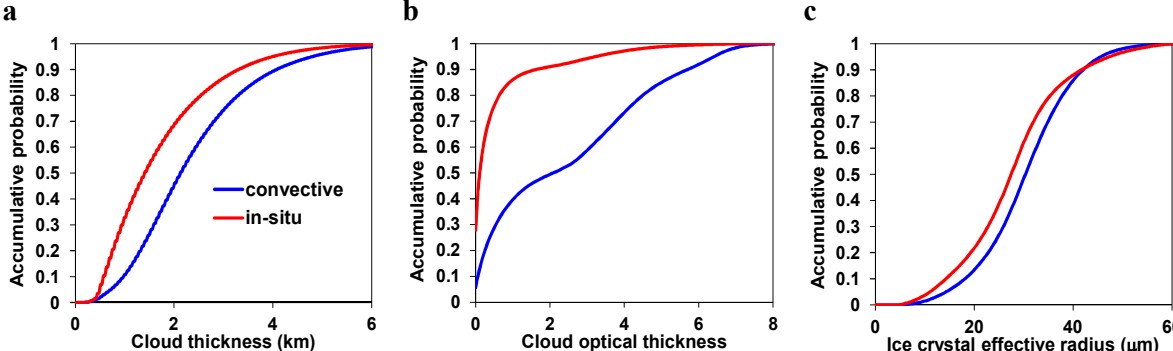

1     Figure 2. Accumulative probability distribution of the properties of two ice cloud types: (a)

2     cloud thickness, (b) cloud optical thickness, and (c) $R_{ei}$.

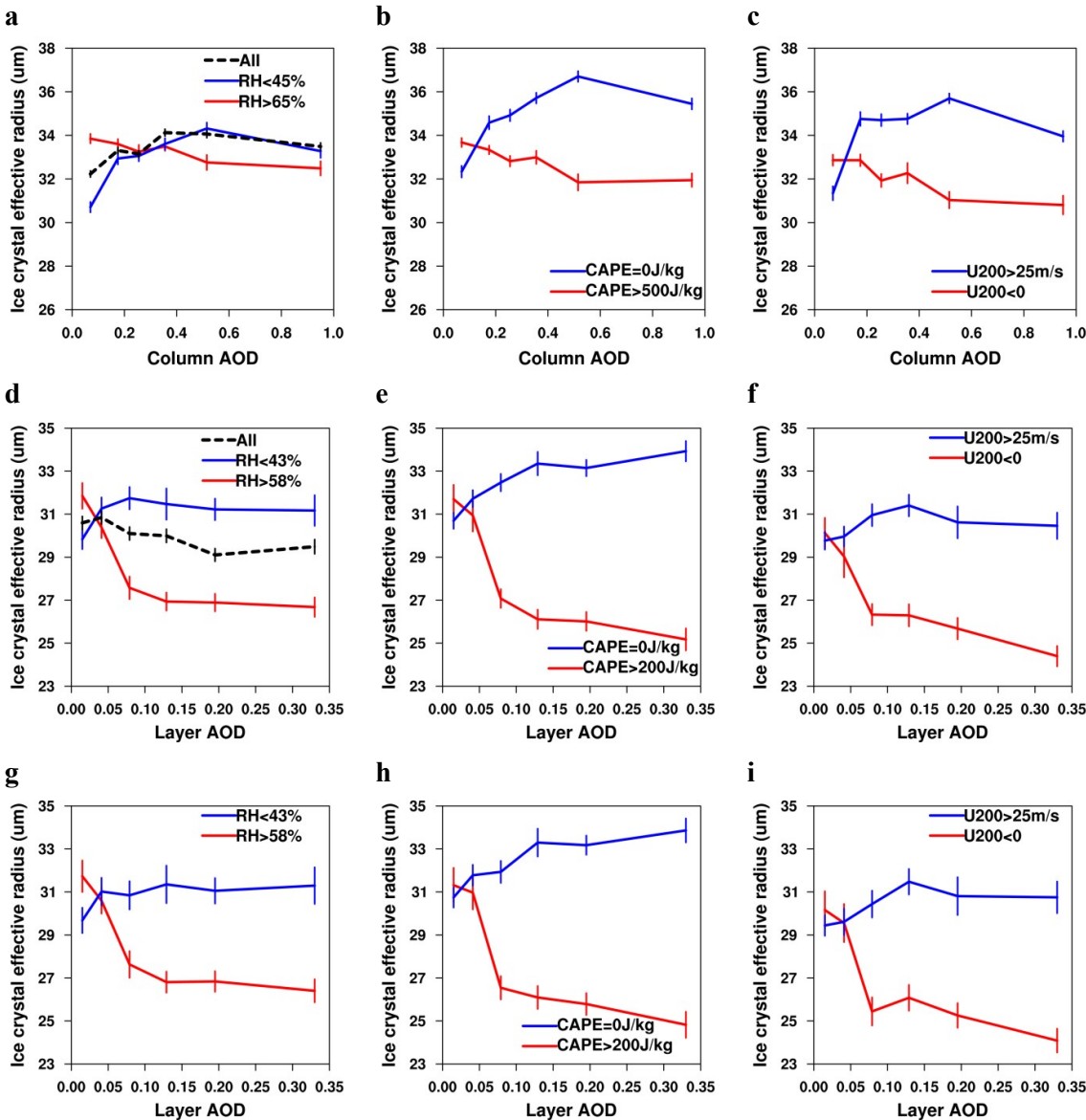

Figure 3. Changes in $R_{ei}$ of convection-generated and in-situ ice clouds with aerosols. (a-c) Changes in $R_{ei}$ of convection-generated ice clouds with AOD for different ranges of (a) $RH_{100-440hPa}$, (b) CAPE, and (c) U200. (d-f) Changes in $R_{ei}$ of in-situ ice clouds with layer AOD for different ranges of (d) $RH_{100-440hPa}$, (e) CAPE, and (f) U200. (g-i) The same as (d-f) but for the profiles with dust aerosols only. The meteorological parameters are divided into 3 ranges containing similar numbers of data points, and the curves for the medium range are not shown. Note that we use column AOD and layer AOD mixed with ice clouds as proxies for aerosols interacting with convection-generated and in-situ ice clouds, respectively. The definition of error bars is the same as in Fig. 1. The total numbers of samples used for convection-generated and in-situ ice clouds are $2.73 \times 10^4$ and $1.09 \times 10^4$, respectively.

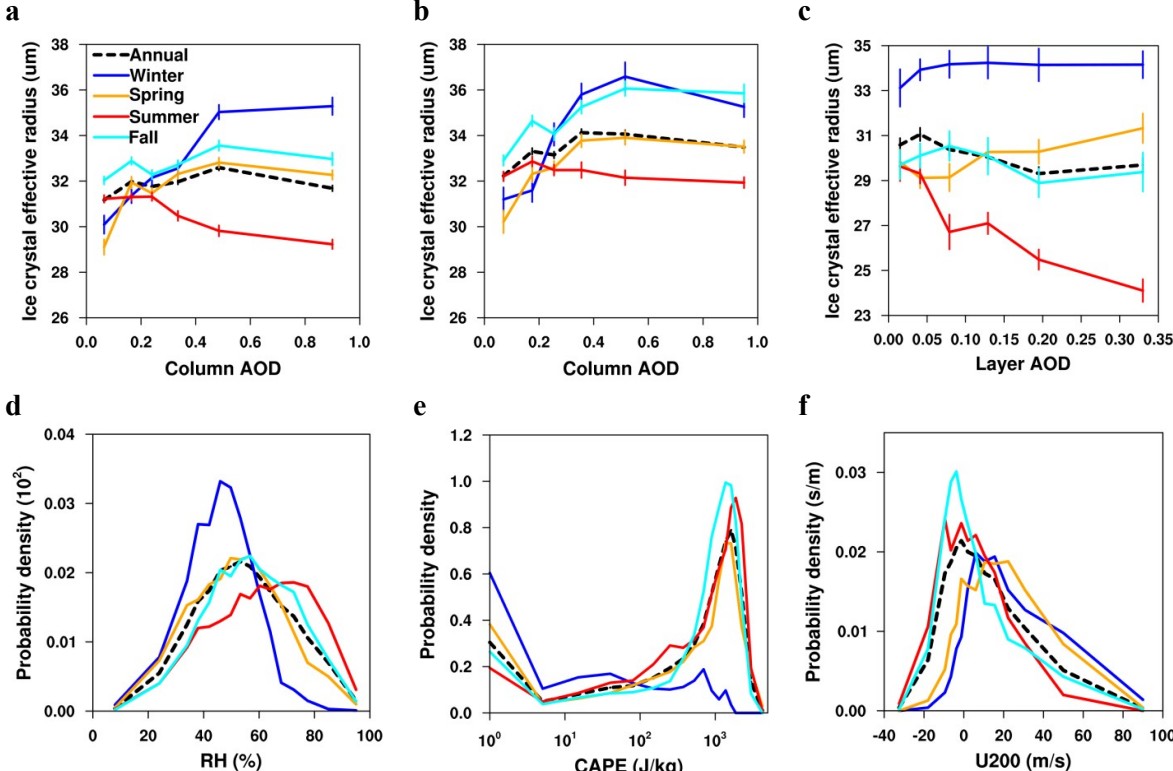

Figure 4. Changes in $R_{ei}$ with AOD and the probability distribution of selected meteorological parameters as a function of season. (a-c) Changes in $R_{ei}$ with AOD as a function of season for (a) all ice clouds, (b) convection-generated ice clouds, and (c) in-situ ice clouds. (d-f) The probability distribution of (d) $RH_{100-440hPa}$, (e) CAPE, and (f) U200 as a function of season. Definitions of season are as follows: Winter – December, January, and February; Spring – March, April, and May; Summer – June, July, and August; Fall – September, October, and November. The definition of error bars is the same as in Fig. 1. The total numbers of samples used in (a, d-f), (b), and (c) are $5.68\times10^4$, $2.73\times10^4$, and $1.09\times10^4$, respectively.

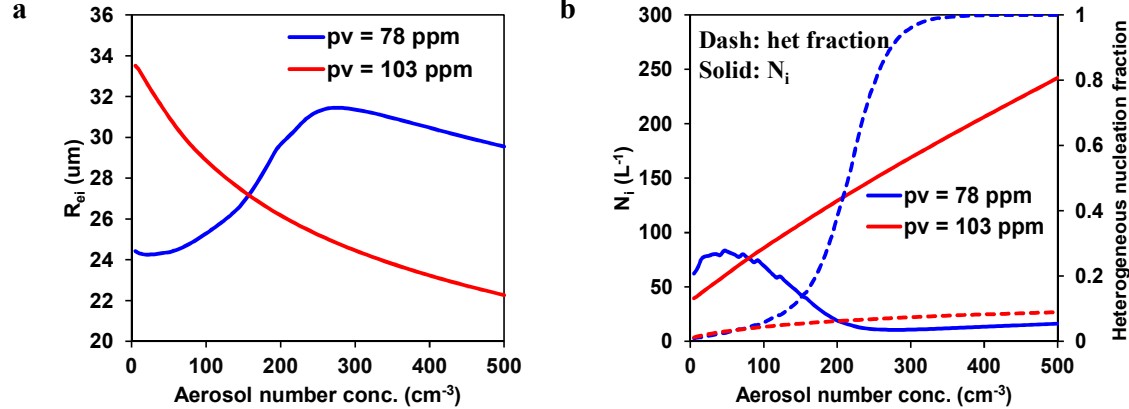

Figure 5. Simulated changes in (a) ice crystal effective radius ($R_{ei}$) and (b) ice crystal number
concentration ($N_i$) and the fraction of ice crystal number produced by heterogeneous
nucleation as a function of the total aerosol number concentration. Simulations are conducted
for two initial water vapor mass mixing ratios (pv), an indicator of available water amount for
ice formation. The ratios of externally mixed dust (deposition INP), coated dust (immersion
INP), and sulfate (not INP) are prescribed with values of 0.75:0.25:10000 in all experiments.