# Peer review of "Impact of aerosols on ice crystal size"

_Atmospheric Chemistry and Physics, 2017_

## Referee Comment (RC1) · Anonymous Referee #1 · 19 Aug 2017

This study investigates the impact of aerosols on ice crystal effective radius (Rei) by using satellite data and parcel model. It reveals the different dependencies of Rei and aerosol optical depth (AOD) under high relative humidity (RH) regime and low RH regime. The mechanisms to cause the difference are discussed and approved by parcel model. The results would help better understand ice cloud microphysical process and better estimation of climate effect of aerosols. In general, the manuscript is well organized. Thus, I suggest a minor revision before publication. The suggestions are list as following:

1. Rei from satellite data retrievals are based on the reflectance of two wavelength (Platnick et al., 2015). Satellite data retrievals need some assumptions and may have some uncertainties. For examples, the surface spectral albedo data is needed to get

the retrievals results. This study focuses on East Asia and surrounding areas, for which most regions are land area. Land surface albedo data may have larger uncertainty, compared with ocean surface albedo. Moreover, a gamma particle size distribution consisting of severely-roughened aggregated column is used in satellite data retrievals (Platnick et al., 2015). Single scattering albedo (SSA) and asymmetric factor needed for retrievals are based on this assumption. Do you think how do these uncertainties affects the results in this study?

2. The particle in ice cloud may have different types and morphologies. For example, in WRF-CHEM, cloud ice, snow, and grapple are used. Platnick et al. (2015) also mentions "solid bullet rosettes" and "solid aggregate plates". Optical properties of each types of particle are quite different. Rei is based on gamma distribution of aggregated column in satellite data retrievals. Thus, the shift of Rei may be caused both by shift of particle size distribution and change of particle type. The types of ice particle formed by homogeneous nucleation and heterogeneous nucleation might be different. Do you think the different type of particle would also be a possible reason, besides the shift of size distributions?

3. There are many small figures in Figure 1, Figure 3, Figure 4 and Figure 5. Some of them are used to support similar conclusions. Maybe the author could consider placing some of them into supplemental information for better understanding of readers.

4. The criteria for low RH and high RH in Figure 1 and Figure 3 are 45% and 65%. But the criteria for Figure 4 is 43% and 58%. Is there any reason for the differences? Will the criteria affect the statistic results?

5. In parcel model results part, water vapor mass mixing ratios and aerosol number concentration are used, which are different from satellite data part, i.e., AOD and RH. Is it possible to use same variables for better comparison?

References

Platnick S., King M. D., Meyer K. G., Wind G., Amarasinghe N., Marchant B., et al. MODIS cloud optical properties: User guide for the Collection 6 Level-2 MOD06/MYD06 product and associated Level-3 Datasets. 2015.

---

## Referee Comment (RC2) · Anonymous Referee #2 · 28 Aug 2017

General Comments:

This referee agrees with the authors that this may be the first paper that studies the impact of aerosol concentration on cirrus cloud microphysics (through ice effective radius Rei in this case). The satellite observations appear valuable to our efforts to understand cirrus cloud-aerosol-radiation interactions, and should ultimately be published in ACP. However, the observations are interpreted narrowly, and much greater scope for interpretation should be provided. Alternate interpretations of the satellite retrievals have been provided under "Major Comments".

While the observations may contribute to our understanding of cirrus cloud-aerosol interactions, this may not be true for the cirrus cloud modeling work presented, as indicated below. It is recommended that Sect. 3.4 be dropped from the paper unless

the concerns listed below can be adequately addressed. That is, the cloud model should predict cloud properties that are representative of in situ cirrus clouds, and the conditions assumed should also be representative. These modeling results are also irrelevant to anvil cirrus clouds, for the reasons stated in (14) below.

The paper is well written and organized, with high quality figures. The observational methodology appears appropriate for this task; the other referee appears to be an expert in this area. The amount of supplementary material appears appropriate.

Major Comments:

1) Page 7, line 24: At what RHi do the deposition INP activate?

2) Section 3: To gain confidence in the reported retrievals of Rei, these Rei retrievals could be compared against another Rei retrieval method reported in the literature. A global analysis of Rei is reported in Hong and Liu (J. Climate, 2015), based on CloudSat-CALIPSO measurements using the "DARDAR" method (a different method than used in this study). Although Hong and Liu do not relate Rei to aerosols, Rei is related to temperature, altitude and cloud optical depth, often as a function of latitude zone and season. Please make some comparisons, as direct as possible, between Hong and Liu Rei values and those reported in this paper.

3) Section 3.1: The error bars in Fig. 1 and elsewhere denote standard errors ($\sigma/\sqrt{N}$) where $\sigma$ is the standard deviation and N is the sample number. This makes the relationships difficult to interpret since we do not know what N is. Please use only $\sigma$ for the error bars so the reader can better evaluate these relationships.

4) Page 9, line 14: Higher RH and CAPE imply that an air parcel will experience a longer time period exceeding ice saturation (i.e. longer time for supersaturation development, increasing the odds of exceeding the RHi threshold for homogeneous ice nucleation (henceforth hom)). This point could be made more clear.

5) Page 10, lines 3-11 (1st paragraph): The similar dependence of Rei on column

[Figure]

AOD (for all aerosol) and column AOD for dust aerosol only is critical to this study, and supports the assumption that ice nuclei (henceforth IN) concentration increases with increasing column AOD. However, this correspondence has only been demonstrated for column AOD and not for layer AOD (where layer AOD corresponds to cirrus cloud levels). Dust is often confined below cirrus cloud levels, and a column AOD-dust AOD relationship does not imply that one exists for layer AOD. Please make this point here.

6) Section 3.2: Rei is positively related to aerosol optical depth (AOD) under relatively dry conditions up to column AOD $\sim$ 0.5 for convective ice clouds and up to $\sim$ 0.13 AOD for in situ ice clouds. These Rei-AOD relationships in Fig. 1, 3 and 4 (for drier conditions) appear to result from competition effects between heterogeneous ice nucleation (henceforth het) and hom, where hom prevails at low AOD and het prevails at higher AOD. As het overtakes hom, Rei increases and ice crystal number concentration, Ni, decreases. This is known as the negative Twomey effect as first described by Kärcher and Lohmann (2003, JGR). Please explain this more thoroughly, citing this paper.

7) Section 3.2: Please state what percentage of the samples were convective vs. in situ.

8) Page 10, lines 32-33: For moist conditions in Fig. 3, this decrease in Rei with increasing AOD is no more than 2 microns, and the error bars show $\sigma/\sqrt{N}$, not $\sigma$ ($\sigma$ should be shown for meaningful interpretation). It is hard to argue that a significant decrease in Rei has occurred with increasing AOD.

More discussion is needed here. As described in Kramer et al. (2016, ACP) and Luebke et al. (2017, ACP), anvil cirrus are a type of "liquid origin cirrus" where liquid cloud droplets contribute to the ice phase as they vertically advect into the cirrus zone (T < 235 K), freezing as they enter this zone. Ice particles from lower levels can also advect into the cirrus zone, especially for anvil cirrus. Cirrus ice from both sources can be viewed as "pre-existing ice" from a nucleation purview, which provides considerable ice surface area that suppresses the increase of ice supersaturation and prevents the

RHi threshold for hom from being attained (Shi et al., 2015, ACP). For this reason, any new ice crystals formed in anvil cirrus are generally expected to form through het or ice crystal multiplication processes. This appears valid for both drier and moist conditions.

Lawson et al. (2015, JAS) combine laboratory measurements, in situ observations and modeling to show that Ni in tropical, convective cumulus clouds is dominated by ice multiplication, which may explain the relatively flat behavior of Rei for high CAPE, high RH and negative U. For such moist conditions, cloud droplets may grow to larger sizes required for ice multiplication. Ice crystals produced this way may be advected by the updraft into the anvil cirrus.

For zero CAPE, lower RH and positive U, ice multiplication may be less important (due to smaller droplet sizes), allowing Rei to increase with increasing AOD, characteristic of hom being overtaken by het (negative Twomey effect). For AOD > 0.4, Rei decreases in accord with het and increasing IN (positive Twomey effect expected when het prevails).

Please expand your discussion to include these points when discussing Fig. 3.

9) Section 3.2, Fig. 4: For AOD < 0.10, the in situ cirrus Rei behavior for lower RH, zero CAPE and positive U could be interpreted as a negative Twomey effect with het overtaking hom due to increasing IN. For AOD > 0.10, if IN conc. is proportional to AOD, the trend should reverse with Rei decreasing with increasing AOD. This does not occur, and there is no evidence that the layer AOD is proportional to dust conc. as noted earlier. Thus it is possible that IN concentration is not tracking the layer AOD, and that IN conc. is relatively constant with AOD. This might explain the relatively flat Rei behavior for AOD > 0.10. Please point this out in the paper.

For the in situ cirrus Rei behavior for higher RH, higher CAPE and negative U (red curves), the interpretation given in this paper makes some sense. The freezing of solution droplets (i.e. hom) may be largely responsible for the decrease in Rei with increasing layer AOD.

10) Page 11, lines 23-26: As stated in the paper, convective clouds vertically advect ice formed via het across the -35 C isotherm, but this "pre-existing ice" greatly suppresses supersaturations and generally prevents the RHi from reaching the RHi threshold for hom (Shi et al., 2015, ACP). This may be true even for the "moist" convective conditions. Please include these points in the discussion (Sect. 3.2).

11) Page 11, lines 29-32: Please state what percentage of sampled clouds were convective vs. in situ for each season. This is important for understanding the regional radiative implications of this work.

12) Section 3.3, Fig. 5b: As noted under (8), ice multiplication can explain the relatively flat behavior of Rei during summer, and perhaps for spring and fall for AOD > 0.4. During winter, CAPE is much lower (see Fig. 5e), suggesting ice multiplication is less important here and Rei decreases for AOD > 0.4 in accord with het and increasing IN. For AOD < 0.4 during winter, spring and fall, Rei increases with increasing AOD, characteristic of hom being overtaken by het. (neg. Twomey effect). Please note this in the paper in regards to Fig. 5b.

13) Section 3.3, Fig. 5c: The summer in situ cirrus Rei behavior could be interpreted as a Twomey effect resulting from het and increasing IN, where deep convection injects more IN into the upper troposphere, thus promoting het. The deep convection during summer promotes tropospheric mixing, making it more likely that IN concentrations at cirrus levels track the layer AOD. It could also be argued that the flat in situ behavior during other seasons could be an indication that IN concentration is not tracking the layer AOD, and that IN concentration is relatively constant with AOD (otherwise, an initial increase in Rei should be followed by a decrease in Rei as AOD increases). The different Rei values could then be explained in terms of seasonal differences in IN concentration, with lowest IN concentration in winter and highest in summer. Please discuss these points in the paper.

14) Page 13, lines 11-14: These modeling results may not apply to anvil cirrus for the

reasons stated in (8) and (10). That is, Ni and Rei in anvil cirrus may be dominated by het and ice multiplication processes. Pre-existing ice should suppress RHi, suppressing hom, making the modeling results irrelevant to anvil cirrus.

15) Page 13, lines 23-28: The modeled Rei values for in situ cirrus clouds are $\sim$ 1/3 those retrieved in this study for such clouds (and are typically $\sim$ 1/3 or less of those from aircraft sampling of in situ cirrus clouds; e.g. Mishra et al., 2014, JGR). For a 30 minute simulation time, the predicted values appear unrealistic. Isometric ice crystals grown at -22 °C reach $\sim$ 100 microns in size after 10 minutes (Takahashi et al., 1991, J. Meteor. Soc. Japan), and would be much larger had the growth times been extended to 30 minutes. While growth rates will be lower at cirrus cloud temperatures, and vapor competition effects can limit growth rates, 30 minutes of growth time should still produce Rei values typical of cirrus clouds, which typically range from 10 and 45 $\mu$m at cirrus cloud temperatures based on aircraft measurements (Mishra et al., 2014, JGR).

The small Rei values imply very high Ni (assuming typical IWCs). Please also plot Ni vs. aerosol number conc. and comment on the realism of the Ni and IWC values.

The text here states variable updraft velocities as a possible reason for the small Rei predicted, but Sect. 2.3 states that a constant updraft velocity (w) of 0.5 m/s is applied throughout the 30 minute simulation time. The parcel model here is simulating in situ cirrus clouds, and w = 0.5 m/s is very high and not representative for in situ cirrus clouds. Hom is most sensitive to the cooling rate or w, and this simulation strongly favors hom due to the high w assumed. Hom can partly explain the small Rei values, but only when hom dominates. It cannot explain the black curve in Fig. 6a where het dominates for aerosol conc. above 200 cm-3; Rei should be $\sim$ 3 times larger here. To summarize, the simulation here is not representative of in situ cirrus clouds and thus should not be used to interpret the satellite measurements.

16) Section 3.4, Fig. 6a: The beginning of the black curve is a manifestation of the "negative Twomy effect" (Karcher & Lohmann 2003, JGR) as hom is overtaken by het.

The slope should become negative after aerosol conc. exceeds 200 cm-3 as increasing IN increases Ni, reducing Rei, but this does not happen. It is not clear why Rei does not decrease in this region.

17) Section 3.4, Fig. 6b: As per my understanding, the initial water vapor mass mixing ratio (pv) determines the level of condensation and thus the portion of the 30 min. simulation time available for supersaturation development. In general, the INP concentrations assumed are sufficiently low to allow attainment of the hom RHi threshold, except for the pv = 38 ppm simulation which has less time for supersaturation development. If this is correct, then please make this clear in the text for greater clarity among the readership. In general, if this modeling section can be made relevant to in situ cirrus clouds, it needs to be expanded and explained better.

18) The following reference: "Ikawa, M., and Saito, K.: Description of a Non-hydrostatic Model Developed at the Forecast 38 Research Department of the MRI, Meteorological Research Institute, Tsukuba, Ibaraki, 39 Japan, 1991." is unconventional, and I wonder whether this is readily accessible. Can it be improved?

Minor Comments:

1) Page 4, line 26: This might be a good place to state that your samples are strictly single-layer ice clouds, instead of at the end of this paragraph.

2) Page 5, line 1: Does cloud type assignment depend exclusively on the way it is flagged?

3) Page 9, lines 15-16: Please indicate that U is the zonal wind as opposed to the meridional wind, and that positive U implies westerly winds; negative U implies easterly winds.

4) Page 12, line 32: Poor sentence; fix grammar. Should say something like "formation of more numerous and smaller ice crystals."

5) Page 13, line 15: Suggest replacing "discrepant" with "different" here and elsewhere

throughout the paper.

---

## Author Comment (AC1) · 8 Nov 2017

We thank the reviewer for the valuable comments. We have followed these comments in revising the manuscript. Please find our point-to-point responses and revised manuscript in the attachment.

Please also note the supplement to this comment: https://www.atmos-chem-phys-discuss.net/acp-2017-548/acp-2017-548-AC1-supplement.zip

---

## Author Response (AR1)

**Reviewer 1**

This study investigates the impact of aerosols on ice crystal effective radius (Rei) by using satellite data and parcel model. It reveals the different dependencies of Rei and aerosol optical depth (AOD) under high relative humidity (RH) regime and low RH regime. The mechanisms to cause the difference are discussed and approved by parcel model. The results would help better understand ice cloud microphysical process and better estimation of climate effect of aerosols. In general, the manuscript is well organized. Thus, I suggest a minor revision before publication.

Response: We thank the reviewer for the valuable comments. We have followed these comments in revising the manuscript. Point-to-point responses are given below.

**The suggestions are list as following:**

1. Rei from satellite data retrievals are based on the reflectance of two wavelength (Platnick et al., 2015). Satellite data retrievals need some assumptions and may have some uncertainties. For examples, the surface spectral albedo data is needed to get the retrievals results. This study focuses on East Asia and surrounding areas, for which most regions are land area. Land surface albedo data may have larger uncertainty, compared with ocean surface albedo. Moreover, a gamma particle size distribution consisting of severely-roughened aggregated column is used in satellite data retrievals (Platnick et al., 2015). Single scattering albedo (SSA) and asymmetric factor needed for retrievals are based on this assumption. Do you think how do these uncertainties affects the results in this study?

**Response**: Thank you for the comments. The MODIS team has performed a comprehensive assessment of the pixel-level uncertainty in  $R_{ei}$  retrievals, which has been incorporated in the Collection 6 Level 2 cloud product (MYD06). This uncertainty evaluation takes into account a variety of error sources, including 1) instrument calibration, 2) atmospheric corrections, 3) surface spectral reflectance, and 4) forward radiative transfer model, e.g., the size distribution assumption (Platnick et al., 2015). The pixel-level  $R_{ei}$  uncertainties for the samples used in this study are 6.41% ± 4.97% (standard deviation). We used mean  $R_{ei}$  within certain AOD bins and the uncertainties are smaller than those for individual pixels. Also, we focus on  $R_{ei}$  changes in response to aerosol loading instead of absolute  $R_{ei}$  values. For these reasons, the  $R_{ei}$  uncertainty ranges are much smaller than the magnitude of  $R_{ei}$  trends depicted in our study (Figs. 1 and 3). We note that the current uncertainty evaluation has not considered the assumptions of ice crystal habit, which will be discussed in the response to the reviewer's second comment.

Following the reviewer's comment, we have added the discussion on the uncertainties of satellite retrieval of  $R_{ei}$  in the revised manuscript. (Page 3, Line 25 to Page 4, Line 4)

2. The particle in ice cloud may have different types and morphologies. For example, in WRF-CHEM, cloud ice, snow, and grapple are used. Platnick et al. (2015) also mentions "solid bullet rosettes" and "solid aggregate plates". Optical properties of each types of particle are quite different. Rei is based on gamma distribution of aggregated column in satellite data retrievals. Thus, the shift of Rei may be caused both by shift of particle size distribution and change of particle type. The types of ice particle formed by homogeneous nucleation and heterogeneous nucleation might be different. Do you think the different type of particle would also be a possible reason, besides the shift of size distributions?

**Response**: We thank the reviewer for this valuable comment. Based on previous studies (Bailey and Hallett, 2009; Lawson et al., 2006; Lynch et al., 2002), the habit of ice crystals is dependent on a number of factors, among which the most important one is temperature, followed by ice supersaturation ratio. In this study we focus on Rei changes with aerosol loading, for which temperature does not appear to have noticeable effect. For supersaturation ratio, the formation of ice crystals under moist conditions (high RH, high CAPE, or negative U200) is dominated by homogeneous nucleation, therefore the ice supersaturation ratio surrounding ice crystals is usually very low and the ice habit is not likely to change significantly with aerosol loading. Under drier conditions (low RH, low CAPE, or positive U200), however, heterogeneous nucleation gradually takes over homogeneous nucleation with aerosol loading increase. Subsequently, the supersaturation ratio surrounding ice crystals would become higher, possibly leading to changes in ice crystal habit. Considering that a single habit (i.e., aggregated column) is assumed in the satellite retrieval algorithm, ice habit changes could possibly induce changes in the satellite-retrieved Rei. However, this retrieval bias should not change our major conclusion about the aerosol impact on ice crystal size, which has been supported by the cloud parcel modeling used in this study.

It should be noted that satellite remote sensing of ice clouds focuses on bulk (averaged) quantity and it is apparent that a single complex rough aggregate shape gives a more consistent retrieval from different MODIS bands and has been adopted for the objective of global ice cloud retrieval. We respectfully submit that at the present time space remote sensing does not have the capability to differentiate ice crystal shapes. The quantitative assessment of the impact of ice crystal habit on satellite retrievals of  $R_{ei}$  is a very complicated and difficult task that merits further in-depth study. We have added these discussions in the revised manuscript. (Page 17, Line 13-30)

3. There are many small figures in Figure 1, Figure 3, Figure 4 and Figure 5. Some of them are used to support similar conclusions. Maybe the author could consider placing some of them into supplemental information for better understanding of readers.

**Response**: Following the reviewer's comment, we have moved some panels of the original Figs. 3 and 4 to the Supplementary Information (Fig. S4 in the revised manuscript). The remaining

panels of these two figures are combined into Fig. 3 in the revised manuscript. For the analysis of the impact of different meteorological parameters, we would prefer to keep the current layout after careful consideration for two reasons. First, the key conclusion of water vapor modulation needs to be supported by the analysis with respect to multiple meteorological parameters, including RH, CAPE, and U200, instead of a single parameter. Second, it may be more convenient to the readers to put these figures in the main text so that they do not need to frequently switch between the main text and Supplementary Information.

4. The criteria for low RH and high RH in Figure 1 and Figure 3 are 45% and 65%. But the criteria for Figure 4 is 43% and 58%. Is there any reason for the differences? Will the criteria affect the statistic results?

**Response**: The probability distributions of RH (as well as other meteorological parameters) are different for convection-generated and in-situ ice clouds. We used different thresholds so that there are approximately the same samples in each meteorological range. We have also tried to apply the same breaking points for both ice cloud types, and found that the  $R_{ei}$ -aerosol relation patterns are retained, but the error bars are larger for some meteorological ranges containing fewer samples.

5. In parcel model results part, water vapor mass mixing ratios and aerosol number concentration are used, which are different from satellite data part, i.e., AOD and RH. Is it possible to use same variables for better comparison?

**Response**: The reviewer's point is well taken. However, we submit that it is a difficult task to undertake a comprehensive comparison unless a more detailed 3D model is used. In satellite data analysis, we used column/layer AOD and RH averaged between 100-440 hPa (or CAPE, U200) as proxies for aerosol loading related to ice clouds and overall available water amount at the upper atmosphere, respectively. However, the cloud parcel model only tracks the aerosol number concentration and water vapor within a single air parcel. It is clear that a direct and quantitative comparison between satellite observations and model results requires developing a 3-D atmospheric model and analysis, a difficult task for further investigation in the future.

Although the indices are not exactly the same, we submit that the simulated dependency of  $R_{ei}$  on aerosols could be used to qualitatively interpret the observed relationships, because the indices used in satellite analysis (AOD and RH averaged between 100-440 hPa) and parcel model (aerosol number concentration and water vapor mixing ratio) are closely correlated with each other, and that the meteorological parameters and aerosol concentration ranges used in the simulations are representative of typical in-situ ice clouds.

We have included these discussions in the revised manuscript. (Page 16, Line 32 to Page 17, Line 12)

**References**

Platnick S., King M. D., Meyer K. G., Wind G., Amarasinghe N., Marchant B., et al. MODIS cloud optical properties: User guide for the Collection 6 Level-2 MOD06/MYD06 product and associated Level-3 Datasets. 2015.

**Reviewer 2**

General Comments:

This referee agrees with the authors that this may be the first paper that studies the impact of aerosol concentration on cirrus cloud microphysics (through ice effective radius Rei in this case). The satellite observations appear valuable to our efforts to understand cirrus cloud-aerosol-radiation interactions, and should ultimately be published in ACP. However, the observations are interpreted narrowly, and much greater scope for interpretation should be provided. Alternate interpretations of the satellite retrievals have been provided under "Major Comments".

While the observations may contribute to our understanding of cirrus cloud-aerosol interactions, this may not be true for the cirrus cloud modeling work presented, as indicated below. It is recommended that Sect. 3.4 be dropped from the paper unless the concerns listed below can be adequately addressed. That is, the cloud model should predict cloud properties that are representative of in situ cirrus clouds, and the conditions assumed should also be representative. These modeling results are also irrelevant to anvil cirrus clouds, for the reasons stated in (14) below. The paper is well written and organized, with high quality figures. The observational methodology appears appropriate for this task; the other referee appears to be an expert in this area. The amount of supplementary material appears appropriate.

**Response:** We thank the Reviewer for constructive comments and suggestions. We have followed them carefully in revising the manuscript.

In particular, we have interpreted satellite observations more broadly (see our responses to the reviewer's  $5^{\text{th}} - 13^{\text{rd}}$  major comments). We have also substantially improved the model simulations for application to in-situ ice clouds and clarified that the modeling results are not applicable to convection-generated ice clouds (see our responses to the reviewer's  $14^{\text{th}} - 16^{\text{th}}$  major comments).

Below is a point-by-point response.

Major Comments:

1) Page 7, line 24: At what RHi do the deposition INP activate?

Response: Thank you. We have added the following sentence in the text. (Page 7, Line 19-22)

"The deposition nucleation on externally mixed dust (deposition INP) and immersion nucleation of coated dust (immersion INP) are parameterized following the work of Kuebbeler et al. (2014); the critical ice supersaturation ratios are 10% (T  $\leq$  220 K) or 20% (T > 220 K) for the former, and 30% for the latter."

2) Section 3: To gain confidence in the reported retrievals of Rei, these Rei retrievals could be compared against another Rei retrieval method reported in the literature. A global analysis of Rei is reported in Hong and Liu (J. Climate, 2015), based on CloudSat-CALIPSO measurements using the "DARDAR" method (a different method than used in this study). Although Hong and Liu do not relate Rei to aerosols, Rei is related to temperature, altitude and cloud optical depth, often as a function of latitude zone and season. Please make some comparisons, as direct as possible, between Hong and Liu Rei values and those reported in this paper.

**Response**: Thank you. Stein et al. (2011) has systematically compared the DARDAR  $R_{ei}$  retrievals with the MODIS data, as shown in Fig. R1. The default DARDAR retrievals of  $R_{ei}$  (denoted by VarCloud-OA, left panel) are mostly larger than MODIS's values. This discrepancy is partly induced by different assumptions of ice crystal habit (shape) in these two products. When the DARDAR retrievals are adjusted to mimic the MODIS assumption of ice crystal habit (VarCloud-BR, right panel), the joint distribution of individual  $R_{ei}$  retrievals has its peak close to the ratio of 1 between the two products, indicating a much better agreement. Nevertheless, the overall shape of the distributions indicates that the MODIS retrievals mostly lie between 10 and 50 µm, while both DARDAR products regularly retrieve  $R_{ei}$  above 60 µm. Hong and Liu (2015) reveals that the large  $R_{ei}$  values in DARDAR retrievals are predominantly associated with large cloud optical thickness (> 3.0, particularly > 20). In this study, however, we focus on ice-only clouds (mostly cirrus clouds), which typically have an optical thickness less than 5.0 (see Fig. 2 in the main text). For this reason, the agreement in  $R_{ei}$  between MODIS and DARDAR could be better for the type of cloud used in our analysis.

We have added the discussions above in the revised manuscript, citing Hong and Liu (2015) and Stein et al. (2011). (Page 4, Line 4-17)

Figure R1. A comparison between the MODIS retrievals of  $R_{ei}$  and two DARDAR retrievals: left – the default DARDAR retrieval, denoted by VarCloud-OA; right – an adjusted DARDAR retrieval to mimic the MODIS assumption of ice crystal habit, denoted by VarCloud-BR. Data are from October 2008. Dashed lines in the figures indicate the 1:1 ratio. This figure is adapted from Stein et al. (2011). ©American Meteorological Society. Used with permission.

3) Section 3.1: The error bars in Fig. 1 and elsewhere denote standard errors  $(\sigma/\sqrt{N})$  where  $\sigma$  is the standard deviation and N is the sample number. This makes the relationships difficult to interpret since we do not know what N is. Please use only  $\sigma$  for the error bars so the reader can better evaluate these relationships.

**Response**: Thank you. We submit that both standard error and standard deviation are widely used, but with different focuses. Standard deviation describes how spread out a set of measurements is, while standard error indicates how accurate our estimate of the mean is likely to be (McDonald, 2014). There is a probability of 68.3% that the population (true) mean would be within one standard error of the sample mean, and a probability of 95.4% to be within two standard errors (McDonald, 2014).

In this study,  $R_{ei}$  is affected not only by aerosol loading, but also a number of confounding factors such as meteorology, altitude, ice water content, etc. Some of these factors (e.g., meteorological conditions) may exert even larger effects on  $R_{ei}$  than aerosols. If standard deviations are plotted, we may not gain an idea whether changes in aerosol loading would induce significant changes in mean  $R_{ei}$ . However, the usage of standard errors could highlight the aerosol effects, because the population (true) mean for a given aerosol bin would very likely (with a 68.3% probability) fall within the error bars. Moreover, if the 95% confidence intervals (1.96 × standard error) of  $R_{ei}$  for two aerosol bins do not overlap, we would be sure that mean  $R_{ei}$  for these two aerosol bins are significantly different at the 0.05 level (McDonald, 2014). For these reasons, we submit that the standard error, which has been adopted by many observational

studies on aerosol-cloud interactions (e.g., Jiang et al., 2011; Su et al., 2011; Koren et al., 2010; Li et al., 2011; Wang et al., 2015), appears to be suitable in our study. Additionally, we have specified the total number of samples used in each figure in the revised figure captions.

4) Page 9, line 14: Higher RH and CAPE imply that an air parcel will experience a longer time period exceeding ice saturation (i.e. longer time for supersaturation development, increasing the odds of exceeding the RHi threshold for homogeneous ice nucleation (henceforth hom)). This point could be made more clear.

**Response**: This suggestion is well taken. We have added this point in the revised manuscript:

"Under moist conditions (high RH, high CAPE, or negative U200), an air parcel could experience longer time for supersaturation development, increasing the odds of exceeding the supersaturation threshold for homogeneous ice nucleation." (Page 10, Line 29-32)

5) Page 10, lines 3-11 (1st paragraph): The similar dependence of Rei on column AOD (for all aerosol) and column AOD for dust aerosol only is critical to this study, and supports the assumption that ice nuclei (henceforth IN) concentration increases with increasing column AOD. However, this correspondence has only been demonstrated for column AOD and not for layer AOD (where layer AOD corresponds to cirrus cloud levels). Dust is often confined below cirrus cloud levels, and a column AOD-dust AOD relationship does not imply that one exists for layer AOD. Please make this point here.

**Response**: We have conducted a similar analysis for in-situ ice clouds and layer AOD for which the results are illustrated in Fig. R2 below (Fig. 3 in the revised manuscript). Similar to column AOD, the dependences of  $R_{ei}$  on layer AOD for all aerosols (Fig. R2a-c) and for dust only (Fig. R2d-f) are also similar. Since specific components of dust aerosols have been known as effective INPs, the similar  $R_{ei}$ -layer AOD relations imply that INP concentrations are also positively correlated with layer AOD, and that the proposed mechanisms for water vapor modulation is applicable to in-situ ice clouds and layer AOD.

We have supplemented this analysis in the revised manuscript. (Fig. 3; Page 11, Line 16-21)

---

## Referee Report (RR1)

2nd Review of ACPD article now titled: *Impact of aerosols on ice crystal size*

Author(s): Bin Zhao et al.

MS No.: acp-2017-548

MS Type: Research article

**General Comments:**

I have gone over all of the author responses to my comments, and am happy to say that all have been addressed very well.  The authors have done an excellent job at revising their paper, which reflects a considerable amount of additional work (e.g. the improvements on modeling).  As noted in the first review, this appears to be the first combined observational-modeling study showing the impact of aerosols on cirrus cloud microphysics under various meteorological conditions.  The conceptual framework of the paper is now greatly improved, and I suspect this paper will be widely referenced in regards to cirrus cloud-aerosol-radiation interactions.  There are just a few outstanding issues that should be addressed, as described below.  Congratulations to the authors for this significant advancement in our understanding.

**Potential highlight paper**

This paper may be of particular interest to the broad geoscience community as it appears to be the first combined observational-modeling study to demonstrate how aerosol loading is likely to affect cirrus clouds under various meteorological conditions.  This study should be a valuable resource for climate modelers attempting to predict the SW and LW forcing by cirrus clouds in relation to changes in aerosol loading.

**Major Comments:**

1) Page 8, line 33:  Different investigators use different formula for calculating Rei; which equation is used here?

2) Page 16, lines 11-12, and Fig. 5 in general:  Since the model has been improved to more closely mimic natural processes now (and predicted and observed Rei agree fairly well), it would be interesting to know what fraction of INP result in an ice crystal for aerosol number concentration > 300 $cm^{-3}$ (when nearly all ice crystals are produced by heterogeneous ice nucleation).  This information would be useful for comparing with other cirrus cloud modeling studies.

Combining deposition and immersion INP together, the INP to aerosol ratio is 1:10,000. For an aerosol concentration of 300 $cm^{-3}$, the combined INP concentration should be 0.03 $cm^{-3}$, or 30 $L^{-1}$. But Fig. 5b relates this INP concentration to an ice crystal concentration ($N_i$) of ~ 300 $L^{-1}$. It seems that either I have made a mistake in this calculation (or assumed something incorrect), or the combined INP-to-aerosol ratio should be 1:1000 to account for an $N_i$ of 300 $L^{-1}$ (assuming all INP produce an ice crystal).

**Minor Comments:**

1) Page 4, lines 12-13: Might be more accurate to say that the DARDAR Rei retrievals, corrected for the crystal habit assumption used here, lie mostly between 10 and 80 microns.

2) Page 12, line 29: dcreases => decreases

---

## Author Response (AR2)

General Comments:

I have gone over all of the author responses to my comments, and am happy to say that all have been addressed very well. The authors have done an excellent job at revising their paper, which reflects a considerable amount of additional work (e.g. the improvements on modeling). As noted in the first review, this appears to be the first combined observational-modeling study showing the impact of aerosols on cirrus cloud microphysics under various meteorological conditions. The conceptual framework of the paper is now greatly improved, and I suspect this paper will be widely referenced in regards to cirrus cloud-aerosol-radiation interactions. There are just a few outstanding issues that should be addressed, as described below. Congratulations to the authors for this significant advancement in our understanding.

Response: We thank the reviewer for the in-depth review and detailed suggestions. We have followed these comments in revising the manuscript. Point-to-point responses are given below.

Major Comments:

1)      Page 8, line 33: Different investigators use different formula for calculating Rei; which equation is used here?

Response: We have revised the sentence as follows:

The $N_i$ for a given aerosol number concentration (i.e., a sub-group of experiments) is calculated using an arithmetical mean of the 100 experiments, while $R_{ei}$ is calculated from mean $N_i$ and mean ice volume: $R_{ei}$ = (mean volume/mean $N_i$ * $3/4\pi)^{1/3}$. (Page 8 Line 32 to Page 9 Line 2 in the revised manuscript)

2)      Page 16, lines 11-12, and Fig. 5 in general: Since the model has been improved to more closely mimic natural processes now (and predicted and observed Rei agree fairly well), it would be interesting to know what fraction of INP result in an ice crystal for aerosol number concentration > 300 cm-3 (when nearly all ice crystals are produced by heterogeneous ice nucleation). This information would be useful for comparing with other cirrus cloud modeling studies.

Response: All INPs are consumed to produce ice crystals in this aerosol number concentration range (300-500 cm$^{-3}$). (Page 16, Line 23-26)

Combining deposition and immersion INP together, the INP to aerosol ratio is 1:10,000. For an aerosol concentration of 300 cm-3, the combined INP concentration should be 0.03 cm-3, or 30 L-1. But Fig. 5b relates this INP concentration to an ice crystal concentration (Ni) of ~ 300 L-1. It seems that either I have made a mistake in this calculation (or assumed something incorrect), or the combined INP-to-aerosol ratio should be 1:1000 to account for an Ni of 300 L-1 (assuming all INP produce an ice crystal).

Response: We apologize that we forgot to indicate that in Fig. 5b, $N_i$ is denoted by solid lines while the fraction of ice crystal number produced by heterogeneous nucleation is denoted by dash lines. Therefore,

for an aerosol concentration of 300 cm$^{-3}$ at pv = 78 ppm, $N_i$ is less than 30 L$^{-1}$. It is not exactly 30 L$^{-1}$ because of sedimentation. We have clarified the meanings of solid and dash lines in the corrected figure shown below.

[Figure]

Figure 5. Simulated changes in (a) ice crystal effective radius ($R_{ei}$) and (b) ice crystal number concentration ($N_i$) and the fraction of ice crystal number produced by heterogeneous nucleation as a function of the total aerosol number concentration. Simulations are conducted for two initial water vapor mass mixing ratios (pv), an indicator of available water amount for ice formation. The ratios of externally mixed dust (deposition INP), coated dust (immersion INP), and sulfate (not INP) are prescribed with values of 0.75:0.25:10000 in all experiments.

Minor Comments:

1)      Page 4, lines 12-13: Might be more accurate to say that the DARDAR Rei retrievals, corrected for the crystal habit assumption used here, lie mostly between 10 and 80 microns.

Response: Done. Thank you! (Page 4, Line 11-13)

2)      Page 12, line 29: dcreases => decreases

Response: Done. Thank you! (Page 12, Line 32)